# The Effect of Chronotype on Addictive Eating Behavior and BMI among University Students: A Cross-Sectional Study

**DOI:** 10.3390/nu14142907

**Published:** 2022-07-15

**Authors:** Müge Arslan, Nurcan Yabancı Ayhan, Hatice Çolak, Esra Tansu Sarıyer, Ekin Çevik

**Affiliations:** 1Department of Nutrition and Dietetics, Faculty of Health Science, Uskudar University, Istanbul 34662, Turkey; hatice.colak@uskudar.edu.tr (H.Ç.); esratansu.sariyer@uskudar.edu.tr (E.T.S.); ekin.cevik@uskudar.edu.tr (E.Ç.); 2Department of Nutrition and Dietetic, Faculty of Health Science, Ankara University, Ankara 06560, Turkey; nyabanci@gmail.com

**Keywords:** eating behavior, food addiction, chronobiology disorders, chronotype, addictive eating

## Abstract

The aim of this study was to examine the effect of chronotype on addictive eating behavior and BMI in university students. A questionnaire on their sociodemographic characteristics and eating habits, as well as the three-factor eating questionnaire (TFEQ) and the morningness-eveningness questionnaire (MEQ), were given to students at Üsküdar University. Data were analyzed with IBM SPSS 26.0. This study included a total of 850 students, 46.8% of whom were male and 53.2% were female. The mean BMI was 22.94 ± 3.30 kg/m^2^. As the BMI values of students increased, there was a decrease of 25.6% in the TFEQ scores, regardless of gender (s = −0.256; *p* < 0.001), and an increase of 10.6% in their chronotype scores, meaning that the students tended to be morning types (s = 0.106; *p* < 0.01). A significantly negative, very weak correlation was found between the students’ TFEQ and MEQ scores (s = −0.123; *p* < 0.001). The students’ BMI (*p* < 0.01) and TFEQ scores (*p* < 0.001) were affected by their MEQ scores in a statistically significant way. It was found that each 1-point increase in the MEQ score led to a 4.0% increase in the BMI score and a 15.8% decrease in the TFEQ score. It was observed that the morning-type students tended to be obese in terms of their BMI, and these students demonstrated fewer addictive eating behaviors.

## 1. Introduction

The hierarchical association of biological clocks extending between the brain and the body, known as the circadian system, regulates human physiology and behavior. Depending on the circadian rhythm, many physiological and metabolic changes, such as sleep-wake cycles, feeding behavior, body temperature, and hormone levels, can be observed [1].

The preferences of individuals regarding the timing of sleep and other behaviors are defined as chronotypes and are classified according to the tendency of individuals to be morningness or eveningness types [2]. According to this classification, it has been stated that morning chronotypes go to bed early in the evening and wake up early in the morning, and their performance is better in the morning, while evening chronotypes go to bed late at night, have difficulty waking up in the morning, and their performance is better in the afternoons. Among the factors affecting the chronotypes of individuals are genetic characteristics, age, ethnicity, and gender [3].

It has been determined that individuals’ chronotypes affect their eating behaviors, as well as maintaining their physical and mental health; morning- and evening-type individuals, especially, have different meal patterns and eating habits [4]. It is known that the morningness chronotype exhibits healthier eating behaviors than the eveningness chronotype, and they have better control over overeating; hence, they are more likely to be healthy [5]. It was concluded that individuals who go to bed late at night have a higher body mass index (BMI), higher energy intake, and fast-food consumption at dinner or after 8 pm, and also have lower levels of fruit and vegetable consumption [6].

Overeating, refusal to eat, restriction of eating, discarding food without digesting it or irresistible nocturnal eating can be considered disordered eating behaviors [7]. Differences between the chronotypes are associated with eating behaviors, while eating attitudes are affected by psychosocial factors [8].

The university years are critical since students can make their own food choices; therefore, the tendency toward unhealthy food choices and negative eating behaviors may increase. Eating habits may change, regardless of family habits, influenced by friends and other factors during this period [9].

This study aimed to understand the relationship between eating attitude changes and the individual circadian rhythm during the university years, which is an important period in which to establish new nutritional characteristics.

## 2. Materials and Methods

The population of this cross-sectional and descriptive study consisted of undergraduate university students enrolled at Üsküdar University for the 2021–2022 academic year. The sample group of the study was obtained by choosing a simple random sample, while the sample size was calculated using a sample formula with a certain universe. It was calculated that 400 university students would be sufficient as a sample size, with a sampling error of 0.05 and a 95% confidence interval, for the study to be carried out. The research began once the Non-Interventional Research Ethics Committee of Üsküdar University granted permission for the study research, under reference number 61351342/11 January 2022, and it was completed with 850 students who participated voluntarily. Data collection was carried out face to face.

### 2.1. Body Mass Index (BMI)

The body mass index (BMI) is frequently used in obesity classification, as established by the World Health Organization (WHO) for the evaluation of obesity in adults. The BMI is obtained by dividing the body weight (kg) by the square of the height (m^2^). According to the WHO, a BMI value of below 18.5 kg/m^2^ is underweight, a BMI value between 18.5 and 24.99 kg/m^2^ is normal weight, a value between 25.0 and 29.9 kg/m^2^ is overweight, and a value over 30 kg/m^2^ is defined as obesity [10].

### 2.2. Three-Factor Eating Questionnaire (TFEQ)

The original version of the three-factor eating questionnaire (TFEQ) developed in 1985 consists of two parts, three subscales, and a total of 51 questions [11]. In 2008, the original version of the TFEQ was translated into Turkish for the first time; later, in 2015, Kıraç et. al. analyzed the validity and reliability of the Turkish translation of the revised version of the TFEQ to facilitate its applicability [12,13]. The number of questions was reduced to 18 and the revised version was used in various studies [13,14]. Items 1, 7, 13, 14, and 17 measure the disinhibition of eating control; the maximum score to be taken from this subscale is 20, while the minimum score is 5. Items 3, 6, and 10 measure the level of emotional eating; the maximum score to be taken from this subscale is 12, while the minimum score is 3. Items number 2, 11, 12, 15, 16, and 18 measure the level of cognitive restraint in terms of eating; the maximum score to be taken from this subscale is 24, while the minimum score is 6. The level of susceptibility to hunger is measured by questions 4, 5, 8, and 9, and the maximum score to be taken from this subscale is 16, while the minimum score is 4. A high score on any of the sub-scales in the questionnaire indicates that the eating behavior related to that factor is more dominant [13].

### 2.3. Morningness-Eveningness Questionnaire (MEQ)

The MEQ is a Likert-type scale developed by Horne and Östberg in 1976 to determine the chronotype; a reliability study was conducted in Turkey by Pünduk et al. in 2005 [3,15]. The questionnaire consists of a total of 19 questions to determine the sleep and waking time preferences of the individual, the time period in which they are productive during the day, and the time intervals that they prefer for physical and mental activities. According to the total scores obtained from the questionnaire, 3 different circadian classifications are expressed: the “morning type”, in the range of 59–86 points; the “intermediate type”, in the range of 42–58 points; and the “evening type”, in the range of 16–41 points [3].

### 2.4. Statistical Analysis

Descriptive statistics for the categorical variables (demographic characteristics) were presented in terms of frequency and percentage. The conformity of the numerical variables to normal distribution was determined with the Shapiro–Wilk test. The Mann–Whitney U test (U) was used in the comparison of two independent groups that did not have a normal distribution, while the Kruskal–Wallis H-test (H) was used in the comparison of more than two groups. The results of the multiple comparison tests were expressed as letters next to the medians. Examination of the relationships between the scales was determined by the Spearman rank differences’ correlation coefficient. In the interpretation of the correlation coefficient, it was determined as a “very weak correlation, if <0.2”, a “weak correlation between 0.2 and 0.4”, a “moderate correlation between 0.4 and 0.6”, a “high correlation between 0.6 and 0.8”, and “0.8 > very high correlation” [16]. Regression analysis was used to test the effect between the variables. Regression analysis involves the explanation of the relationship between two related variables, a dependent variable, and an independent variable, with mathematical equivalence [17]. In all the calculations and interpretations, the statistical significance level was considered as *p* < 0.05 and the hypotheses were established as bidirectional. Statistical analysis of the data was performed using IBM SPSS Statistics for Windows 26.0 (IBM Corp., Armonk, NY, USA).

## 3. Results

### 3.1. Demographic and Nutritional Findings

The average age of the students was 21.67 ± 2.83 years; in terms of the participants, 46.8% were males and 53.2% were females. The average BMI of the students was 22.94 ± 3.30 kg/m^2^ and the average water consumption was 1.73 ± 0.57 L/day. The students were mostly (67.9%) educated in health sciences but skipped their meals (73.1%). Among the students, 56.0% did not smoke and 80.7% did not drink alcohol. According to the MEQ score, 77.1% of the students had intermediate circadian type classification, 15.1% were a morning type, and 7.8% were an evening type is given in Table 1.

### 3.2. Comparative Analysis of the Demographic and Nutritional Findings with the TFEQ Scores

A comparison of the demographic and nutritional findings of the university students with the TFEQ scores is given in Table 2.

One of the sub-scales of the TFEQ is the “disinhibition of eating control”; the median score of this scale was statistically higher in the female students (14 (5–19)) than in the male students (13 (5–19)) (U = 71,077; *p* < 0.001), higher in the non-smokers (14 (5–20)) than in the smokers (13 (5–20)) (U = 76,475.5; *p* < 0.001), and higher in the non-drinkers (14 (5–20)) than in the drinkers (12 (5–20)) (U = 47,354.5; *p* < 0.01). According to the “disinhibition of eating control” scores, the median score of the intermediate (14 (5–20)) and evening-type (13 (6–20)) students was statistically higher than the morning-type (11 (5–20)) students (H = 37.013; *p* < 0.001). It was found that there was a statistically significant negative and weak correlation between the “disinhibition of eating control” scores of the university students and their BMI (s = −0.288; *p* < 0.001); as the BMI of the university students increased, there was a 28.8% decrease in the “disinhibition of eating control” scores.

Another sub-scale of the TFEQ is “emotional eating”; the median score of this scale was statistically higher in the non-smokers (8 (3–12)) than in the smokers (7 (3–12)) (U = 75,593; *p* < 0.001). When classified according to the circadian rhythm chronotypes, the “emotional eating” median score was statistically higher in the evening- (8 (3–12)) and intermediate-type students (8 (3–12)) than the morning-type (7 (3–12)) students (H = 9.058; *p* < 0.05). It was found that there was a statistically significant negative and weak correlation between the “emotional eating” scores of the university students and their BMI; as the BMI of university students increased, there was a 20.4% decrease in their “emotional eating” scores (s = −0.204; *p* < 0.001).

The “cognitive restraint of eating” sub-scale median scores of the TFEQ were statistically higher in those students who skipped meals (15.5 (7–24)) than in those who did not (15 (6–22)) (U = 60,565; *p* < 0.01) and was statistically higher in the smokers (16 (6–24)) than in the non-smokers (15 (7–24)) (U = 79,814.5; *p* < 0.01). On the other hand, a statistically significant negative and very weak correlation was found between age (s = −0.080; *p* < 0.05), BMI (s = −0.168; *p* < 0.001), and the “cognitive restraint of eating” scores. Accordingly, as the age and BMI of the university students increased, there was an 8.0% and 16.8% decrease in the “cognitive restraint of eating“ scores, respectively.

The last sub-scale of the TFEQ is “susceptibility to hunger”; the median score of this scale was statistically higher in the females (11.5 (4–16)) than in the males (10 (4–16)) (U = 68,672.5; *p* < 0.001), higher in the non-smokers (11 (4–16)) than in the smokers (10 (4–16)) (U = 76,186; *p* < 0.001), and higher in the drinkers (11 (4–16)) than in the non-drinkers (10 (4–16)) (U = 46,560.5; *p* < 0.01). According to the MEQ scores, the “susceptibility to hunger” median score was statistically higher in the evening-type (11.5 (4–16)) and intermediate-type students (11 (4–16)) than in the morning-type (9 (4–16)) students (H = 31.671; *p* < 0.001). It was found that there was a statistically significant negative and weak correlation between the “susceptibility to hunger” scores of the university students and their BMI; as the BMI of the university students increased, there was a 31.4% decrease in the “susceptibility to hunger” scores (s = −0.314; *p* < 0.001).

The MEQ score was evaluated and was statistically higher in the males (52 (19–82)) than in the females (49 (19–80)) (U = 75,386.5; *p* < 0.001), higher in those who skipped meals (51 (19–80)) than in those who did not (50 (19–82)) (U = 61,377.5; *p* < 0.01), higher in the smokers (52 (19–80)) than in the non-smokers (50 (19–82)) (U = 78,734; *p* < 0.01), and higher in the drinkers (53 (19–80)) than in the non-drinkers (50 (19–82)) (U = 45,117; *p* < 0.001). A statistically significant positive and weak correlation was found between the MEQ score and BMI and, as the BMI of the university students increased, there was a 10.6% increase in the MEQ scores (s = 0.106; *p* < 0.01).

### 3.3. Linear Regression Analysis: Total TFEQ Scores, Adjusted for Gender, BMI, and Total MEQ Score

The results of the multiple linear regression analysis to investigate the determinants of the total TFEQ scores are given in Table 3. It can be seen that the results explained 9.6% of the model variance created (F: 31.067, *p* < 0.01, R^2^: 0.096). After adjusting for confounders, the linear regression analysis revealed that regardless of gender, the BMI and MEQ total scores affected the TFEQ total score (ß: −0.267 and ß: −0.108, respectively).

### 3.4. Analysis of Correlation with the Total MEQ and TFEQ Scores

The correlation with the Total MEQ and TFEQ Scores is given in Table 4. It was found that there was a significant negative and very weak correlation between the students’ TFEQ scores and their MEQ scores (s = −0.123; *p* < 0.001) and there was a 12.3% decrease in the MEQ scores as the students’ TFEQ scores increased.

### 3.5. Analysis of the MEQ, BMI, and Total TFEQ Scores

The effect of university students’ MEQ scores on their BMI values and TFEQ scores is given in Table 5. It was found that the students’ MEQ values significantly affected their BMI and TFEQ scores. The MEQ scores (β = 0.040; *t* = 2.746; *p* < 0.01) explained 0.9% of the variance in the BMI values (R^2^ = 0.009; F = 7.539; *p* < 0.01). The students’ MEQ scores (β = –158; *t* = −4.073; *p* < 0.001) explained 1.9% of the variance in the TFEQ scores (R^2^ = 0.019; F = 16.591; *p* < 0.001). When the results were examined, it was found that a 1-unit increase in the MEQ scores of the university students caused an increase of 4.0% in their BMI values and a decrease of 15.8% in their TFEQ scores.

## 4. Discussion

Of the 850 students, with a mean age of 21.67 ± 2.83 years and a mean BMI of 22.94 ± 3.30 kg/m^2^, 46.8% were male and 53.2% were female. With the increase in BMI in the university students, there was a decrease in the TFEQ scores and an increase in the chronotype scores. As the TFEQ score increased, the MEQ decreased. An increase in MEQ score was related to an elevation in the BMI and a decrease in the TFEQ scores.

In this study, the total TFEQ score, and the median scores of uncontrolled eating and susceptibility to hunger were statistically significantly higher in the female students than in the males. Similar to these results, İkiz [18] observed that the TFEQ scores were higher in female participants. However, once uncontrolled eating scores were classified according to gender, the emotional eating scores were higher in the females, although there was no statistically significant difference. In a cross-sectional study conducted with university students, it was found that female students had greater cognitive restraint and higher emotional eating sub-factor scores [19]. While the increase in responsibilities during the transition from school to university toward adulthood is a stressful period for both males and females, it is known that stress triggers emotional eating tendencies and has an effect on eating behaviors [20,21]. Some studies have shown that uncontrolled eating tendencies due to stress and emotional changes, and emotional eating behavior are more common in females [21,22]. Relevant studies in the literature support the findings of this study. Herein, the female students may have been better able to cope with situations that may trigger emotional hunger, such as an adaptation to new friends and lifestyle when moving away from their family environment, and the exam process, which may create stress, such as uncontrolled eating and emotional hunger susceptibility, compared to the males. In this respect, the strength of this study was that in situations where these stress factors may occur, the study contributes to and supports the training given in parallel with this situation, which can reveal the differences in eating attitudes between males and females and may affect attitudes toward eating in obesity prevention programs.

Among the participants in this study, 77.1% had an intermediate-type circadian rhythm, 15.1% were of the morning type, and 7.8% were of the evening type. The results showed that the male students were more likely to be morning types than the females. Beşoluk [23] conducted a study with university students and 18.1% were found to be a morning type, 0.9% were an evening type, and the rest were of an intermediate type. In this study, although the MEQ scores did not differ significantly according to gender, they were still higher in the males. In a more recent study conducted with university students, the MEQ score was statistically higher in the females, with the majority of participants being of an intermediate type [24]. In the study by Rique et al. [25], 27.6% of the students were of the morning type, 51.6% were of the intermediate type, and 20.8% were of the evening type, while no significant difference was observed between MEQ scores according to gender. Although the results in the literature provide mixed results in terms of gender and chronotypes, previous studies have shown that males are more inclined to be evening eaters than females [26]. However, this new evidence also supports the finding that the incidence of the evening type in females increases as their age increases. Recent findings have indicated that this situation may be related to aging-associated biological changes, increased gender equality in cultures, and environmental, psychological, and developmental factors [27,28].

When adjusted for gender, as the BMI values of the university students increased, the total score for the TFEQ, emotional eating, disinhibition of eating control, and cognitive restraint of eating sub-dimension scores decreased. In an earlier cross-sectional study conducted among university students, no correlation was found between BMI and the total TFEQ scores. Moreover, the cognitive restraint of the eating subfactor scores was significantly higher in the underweight group when compared to the obese and normal weight groups [29]. On the other hand, Erkaya et al. [30] reported that as the BMI values increased, it was observed that TFEQ scores and restrictive eating behavior increased, while in another study, no relationship was found between the cognitive restraint of eating and BMI [31]. Contrary to the current study, Gallant et al. [32] found that there was a positive correlation between the BMI and TFEQ scores in adolescents and that, as the score for the cognitive restraint of eating increased, the waist circumference and body fat ratio also tended to increase in the adolescents. In addition, it is obvious that uncontrolled eating behavior is more common in obese individuals due to many factors, including impaired appetite regulation and emotional triggers [31,33,34]. Although the above studies on the relationship between the cognitive restraint of eating and BMI values had mixed results, the decrease in the conscious eating behavior restriction scores in the current study can be explained by the weakening of eating control in the obese students. In this respect, the strength of this current work is that in programs to combat obesity, it is important to support obese individuals to raise awareness by revealing the situations that may occur regarding the eating attitudes of obese individuals in hunger situations.

According to the results of this study, there was a statistically significant increase in the MEQ scores as the BMI values increased. In other cross-sectional studies conducted with university students, it was observed that the BMI levels were higher in the morning types; these studies showed that there was a significant positive relationship between the BMI and MEQ scores, as seen in the current study [35,36]. Contrary to the results herein, in a study conducted among adolescents, evening chronotypes were found to have higher BMI values [37]. Similarly, Sun et al. [38] reported in their study that the BMI was significantly higher in individuals of the evening chronotype; this was also associated with a decrease in sleep quality. Circadian preferences do cause sleep deprivation and a decrease in sleep quality [37]. This situation, which includes these environmental, psychological, and physiological elements, has an impact on food selection and leads to the formation of obesity. Although some results in the literature have indicated that BMI levels are high in evening types, due to unhealthy food choices and sleep deprivation, there are also studies in which no significant relationship was observed between the chronotype and BMI [39]. In a cross-sectional study of university students, the initial weight of the evening-type students was not greater than that of the morning/intermediate-type students; this finding was explained by the idea that the students might not completely engage in evening-related unhealthy behaviors until they leave home and develop a decision-making autonomy [40]. In addition, students may actually put into practice the opposite of their basic circadian preferences for academic and social reasons, and, thus, may seem like evening chronotypes, based on their sleep timing choices. Therefore, decisions about bedtime and wake-up times may not directly coincide with morning–evening preferences [40]. These conditions may also explain the high BMI of the morning types in the current study. In addition, this may be due to the fact that the effect on BMI was not clearly determined, due to the lack of food consumption and physical activity records from the students in this study.

It was found that there was a statistically significant negative correlation between the cognitive restraint of eating scores and the participants’ age. A study conducted with different age groups stated that although there was no significant difference in the conscious food restriction scores between these age groups, a conscious eating restriction situation occurred with a decrease in food intake as the respondent’s age increased [41]. In another recent study, a positive and significant relationship was found between age and conscious food restriction behavior. Accordingly, it can be seen that both males and females consciously reduce their food intake with advancing age [42]. This result of the current study revealed the strength of this study as a method to help university students increase their awareness of nutrition by revealing their feelings, thoughts, and concerns about nutrition with the progression of time, over the years.

Herein, it was found that an increase in the tendency to be a morningness type may be related to a decrease in students’ impaired eating attitudes. Uncontrolled eating, emotional eating, and hunger sensitivity behaviors were found to be lower in students with morning circadian rhythms. As the predisposition of the students to be an evening type increased, their uncontrolled eating, emotional eating, and hunger sensitivity behaviors increased. In addition, as the predisposition of the students toward the morning type increased, the cognitive restraint of eating behavior scores also increased. It was found that students who skipped meals had a higher tendency toward the morning type and greater cognitive restraint in terms of eating behavior. In a study conducted among university students, healthy eating behavior, such as conscious eating, was found to be associated with morning types, while unhealthy eating behavior, such as uncontrolled and emotional eating, was associated with evening types [43]. In another study, greater conscious restriction and lower uncontrolled eating were observed in morning-type people, while food addiction and hunger sensitivity were observed to be greater in evening-type people [44]. Morning-oriented people have more control over their daily habits and have healthier lifestyles. The fact that this behavior is more dominant in morning-oriented people allows them to control their eating behavior more easily; they exhibit less emotional eating and uncontrolled eating behavior. Cognitive restraint is the intention of controlling food intake to maintain/lose body weight [45]. Morning-oriented people have more control over their daily habits and have healthier eating behaviors, such as conscious restriction and cognitive control [46]. Nutrient intake in evening-oriented individuals is associated with higher nutrient consumption in evening meals, less dietary restriction, a lower-quality diet, and unhealthy nutrition in terms of skipping breakfast [47]. Compared to the morning types, it was reported that greater energy intake in the evening results in less successful weight loss [48]. In line with the literature, the current study showed that the students who exhibited conscious restriction eating behaviors and had a predisposition to the morning type exhibited a controlled appetite and lower hunger sensitivity.

This study examined the chronotype and eating attitudes of university students in Turkey. According to our observations, studies on this subject have generally focused on chronotypes and sleep problems. On the other hand, studies examining the relationship between eating attitudes and BMI are limited; the findings herein will contribute to the literature. The large sample size, the balanced sample distribution in terms of gender, and the face-to-face collection of the data by an experienced researcher were also the strengths of this study. In addition, the results of the study show its strength in terms of increasing the level of consciousness of students who develop anxiety regarding obesity or who have wrong information about this issue by revealing the effect of sleep duration, not only on mental and physiological fitness but also on eating attitudes.

However, since this study was cross-sectional, the results represented a certain period. At the same time, the weakness of the study was that food choices according to chronotype were not examined. Examining food choices in future studies will reveal a clearer relationship between the chronotype and eating attitudes, revealing the patterns of food consumption in addition to the BMI.

## 5. Conclusions

In this study, an increase in MEQ score was related to an elevation in the BMI and a decrease in the TFEQ scores. As the predisposition toward the morning chronotype increased, when compared to the evening chronotype, the BMI values likewise increased; this relationship occurred regardless of gender. The morning-oriented students had greater cognitive restraint regarding eating and the evening-oriented students showed greater disinhibition of eating control, emotional eating, and susceptibility to hunger. In parallel with the increase in the disinhibition of eating control, emotional appetite, and emotional eating, individuals may lose their eating control. With increasing calorie intake, students may gain weight, which may lead to the risk of obesity. According to the results of this study, to increase the applicability of nutritional recommendations, it was seen that it should be shaped according to chronotype rather than being uniform. In evening-type individuals, with nutritional recommendations to address the susceptibility to hunger, disinhibition of eating control, and emotional eating behaviors, many chronic diseases that may occur in the future, especially obesity and the problems accompanying it, can be prevented.

## Figures and Tables

**Table 1 nutrients-14-02907-t001:** Descriptive statistics of the demographic and nutritional findings of the university students.

	Male		Female		Total	
	*n*	%	*n*	%	*n*	%
**Nationality** Turkish	325	81.7	383	84.7	708	83.3
Other	73	18.3	69	15.3	142	16.7
**Age** **(** $\bar{X}\pm SS$ **)**	21.64 ± 2.41		21.70 ± 3.15		21.67 ± 2.83	
**BMI (kg/m^2^)**$\;(\bar{X}\pm SS$)	23.80 ± 2.89		22.17 ± 3.45		22.94 ± 3.30	
**Departments** Health Sciences	236	59.3	341	75.4	577	67.9
Other Sciences	162	40.7	111	24.6	273	32.1
**Water** **Consumption** **(L/day)** **(**$\bar{X}\pm SS$**)**	1.75 ± 0.56		1.71 ± 0.58		1.73 ± 0.57	
**Skipping Meals** Yes	302	75.9	319	70.6	621	73.1
No	96	24.1	133	29.4	229	26.9
**Smoking Status** Yes	210	52.8	164	36.3	374	44.0
No	188	47.2	288	63.7	476	56.0
**Drinking Alcohol Status** Yes	105	26.4	59	13.1	164	19.3
No	293	73.6	393	86.9	686	80.7
**Chronotype type** Morning	65	16.3	63	13.9	128	15.1
Intermediate	306	76.9	350	77.5	656	77.1
Evening	27	6.8	39	8.6	66	7.8

**Table 2 nutrients-14-02907-t002:** Comparison of the TFEQ total and sub-scale scores and the MEQ scores of the university students, according to the demographic and nutritional findings.

		TFEQ-1	TFEQ-2	TFEQ-3	TFEQ-4	TFEQ	MEQ
		Median (Min–Max)	Median (Min–Max)	Median (Min–Max)	Median (Min–Max)	Median (Min–Max)	Median (Min–Max)
**Gender** Male		13 (5–19)	7 (3–12)	15 (9–23)	10 (4–16)	45.5 (21–69)	52 (19–82)
Female		14 (5–19)	8 (3–12)	15 (6–24)	11.5 (4–16)	48 (19–69)	49 (19–80)
U		71,077	86,200.5	86,001.5	68,672.5	75,325.5	75,386.5
*p*		**<0.001**	0.289	0.267	**<0.001**	**<0.001**	**<0.001**
**Age** **(** $\bar{X}\pm SS$ **)**	s	−0.013	0.019	−0.080	−0.036	−0.047	−0.006
	*p*	0.702	0.585	**0.019**	0.295	0.169	0.852
**BMI** **(** $\bar{X}\pm SS$ **)**	s	−0.288	−0.204	0.026	−0.314	−0.256	0.106
	*p*	**<0.001**	**<0.001**	0.453	**<0.001**	**<0.001**	**0.002**
**Water** **Consumption** **(L/day)** **(**$\bar{X}\pm SS$**)**	s	0.010	0.040	−0.168	0.017	−0.043	−0.039
	*p*	0.776	0.238	**<0.001**	0.627	0.206	0.255
**Meal skipping** Yes		13 (5–20)	8 (3–12)	15.5 (7–24)	11 (4–16)	47 (19–69)	51 (19–80)
No		14 (5–20)	7 (3–12)	15 (6–22)	11 (4–16)	47 (21–64)	50 (19–82)
U		69,916	71,015.5	60,565	70,515	68,370	61,377.5
*p*		0.707	0.977	**0.001**	0.852	0.389	**0.002**
**Smoking Status** Yes		13 (5–20)	7 (3–12)	16 (6–24)	10 (4–16)	46 (21–69)	52 (19–80)
No		14 (5–20)	8 (3–12)	15 (7–24)	11 (4–16)	47 (19–69)	50 (19–82)
U		76,475.5	75,593	79,814.5	76,186	80,001.5	78,734
*p*		**<0.001**	**<0.001**	**0.009**	**<0.001**	**0.011**	**0.004**
**Drinking Alcohol Status**							
Yes		12 (5–20)	7 (3–12)	15 (8–22)	11 (4–16)	46 (21–64)	53 (19–80)
No		14 (5–20)	8 (3–12)	15 (6–24)	10 (4–16)	47 (19–69)	50 (19–82)
U		47,354.5	53,275.5	51,461.5	46,560.5	50,603.5	45,117
*p*		**0.002**	0.287	0.088	**0.001**	**0.045**	**<0.001**
**Chronotype type**							
Morning		11 a (5–20)	7 a (3–12)	16 (8–22)	9 a (4–16)	43 a (20–67)	-
Intermediate		14 b (5–20)	8 b (3–12)	15 (6–24)	11 b (4–16)	48 b (19–69)	-
Evening		13 b (6–20)	8 b (3–12)	14 (9–21)	11.5 b (4–16)	48 b (24–65)	-
H		37.013	9.058	5.804	31.671	20.345	-
*p*		**<0.001**	**0.011**	0.055	**<0.001**	**<0.001**	-

TFEQ-1: disinhibition of eating control, TFEQ-2: emotional eating, TFEQ-3: cognitive restraint of eating, TFEQ-4: susceptibility to hunger, TFEQ: three-factor eating questionnaire, MEQ: morningness–eveningness questionnaire; U: Mann–Whitney U test; H: Kruskal–Wallis H test; s: Spearman’s rank differences correlation. ***p* < 0.05** is considered statistically significant; a, b: the difference between medians that do not have common letters is significant (*p* < 0.05).

**Table 3 nutrients-14-02907-t003:** Multiple linear regression analyses of the total TFEQ scores, adjusted for gender, BMI, and total MEQ score.

Outcome	Standardized Coefficients			95.0% Confidence Interval for *ß*	
	*ß*	t	*p*-Value	Lower Bound	Upper Bound
**(Constant)**	67.396	21.759	0.001	610.317	730.476
**BMI**	−0.267	1.547	**<0.001**	−0.893	−0.538
**Gender**	0.052	−7.914	0.122	−0.249	20.103
**Total MEQ Score**	−0.108	−3.272	**<0.001**	−0.196	−0.049

Results from the linear regression models (adjusted for gender, BMI, and total MEQ score). Adjusted R^2^ = 9.6%; ***p* < 0.05** is considered statistically significant; BMI: Body Mass Index; MEQ: Morningness–eveningness questionnaire;Constant: It shows the point where the regression line cuts off the y-axis. Returns the average value that y would take when the value of the x argument was 0.

**Table 4 nutrients-14-02907-t004:** Correlation coefficients between the total TFEQ and MEQ scores.

		TFEQ
**MEQ**	**s**	**−0.123**
	** *p* **	**<0.001**

TFEQ: three-factor eating questionnaire; MEQ: morningness–eveningness questionnaire; ***p* < 0.05** is considered statistically significant.

**Table 5 nutrients-14-02907-t005:** Effect of the MEQ scores on the BMI values and the total TFEQ scores.

	Model	*β*	SE	*t*	*p*-Value	F	*p*-Value
**BMI**	(Constant)	20.890	0.754	27.722	**<0.001**	7.539	**0.006**
	MEQ	0.040	0.016	2.746	**0.006**		
	R = 0.094; R^2^ = 0.9%; adjusted R^2^ = 0.8%						
**TFEQ**	(Constant)	54.207	2.008	26.999	**<0.001**	16.591	**<0.001**
	MEQ	−0.158	0.039	−4.073	**<0.001**		
	R = 0.139; R^2^ = 1.9%; Adjusted R^2^ = 1.8%						

TFEQ: three-factor eating questionnaire, MEQ: morningness–eveningness questionnaire; ***p* < 0.05** is considered statistically significant.

## Data Availability

Not applicable.

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
