# Peer review of "The Effect of Chronotype on Addictive Eating Behavior and BMI among University Students: A Cross-Sectional Study"

_nutrients, 2022, doi:10.3390/nu14142907_

Round 1

Reviewer 1 Report

In this manuscript, the authors examined eating behavior, Chronotype, and BMI among college students and presented a cross-sectional study of these actual conditions and their respective relationships. While this topic is interesting and may contribute to improving the health of young people, there are some problems indicated below.

Major comments
Comment 1.
Table 1 shows that men had a higher BMI than women; Table 2 shows that men had a higher MEQ than women and were more prone to morning chronotypes. Although this study concluded that morning chronotypes were associated with increased BMI, it is possible that this simply reflected gender differences in weight. To resolve this issue, sex-adjusted analyses should be performed or analyses should be performed for each sex. Results should be presented adjusted for other potential confounders if possible.

Comment 2.
In the first paragraph of the Discussion, briefly describe the results of this study and their implications. The authors have presented a comparison of the results with previous studies, but if the results of the previous studies did not agree with the results of the present study, please discuss the factors that led to the discrepancy. At the end of the Discussion, list the strengths and limitations of the present study.

Minor comment

For Table 3, it is not necessary to examine the correlation between the TFEQ subscales and the total score. Nor is it necessary to include it in the text.
Just show the correlation coefficient between the total TFEQ score and the MEQ.
In 3.3 Analysis of MEQ and TFEQ-R18 scores, almost all the values in Table 3 are listed again in the text. It is not necessary to list all of the values, but please provide a brief description.

Author Response

We thank to the reviewer for his/her contribution to our manuscript. We will consider his/her comments in our future studies.

Dear Reviewer 1;

All requested corrections have been applied and painted in yellow. Furthermore, the entire text is "proofread" by the "native speaker".

  1. Major comment 1
    • In line with the referee's suggestion, gender-based linear regression analysis was conducted (Table 3). The research concluded that gender had no impact on the relationship between BMI and chronotype. This conclusion has been added to the discussion section
  2. Major comment 2
    • In accordance with the referee's suggestion, a brief description of the resarch findings was added to the first paragraph of the discussion.
    • The discussion was reviewed and the differences in the results of the resaerch were interpreted.
    • The study's limitaitions and strengths are listed.
  3. Minor comment
    • Table 3 has been revised and the analysis of subscales has been removed (You can see it as table 4 in the revised version of the manuscript ).
    • In line with the final version of the table, typo adjustments (findings, discussion) were conducted.

Reviewer 2 Report

This is a comprehensive study of the effects of habitual eating behavior and chronotype on BMI in Turkish university students from 2021 to 2022. A detailed analysis of the TFEQ-R18 scores for 850 students showed the following: The number of morning students is increasing, and while morning students tend to become obese, they do not have addictive eating behaviors. Although it has academic significance in this respect, I feel that there is a problem in how to assemble and interpret the dissertation. Please refer to the comments below.

Major points

1)     This research was conducted at a time when the new coronavirus infection COVID-19 was a problem all over the world, and I think it is very different from the background of the previous research. There is no mention of that point, and I think it is necessary to consider the peculiarities that COVID-19 gives to this study.

2)     The explanation given between lines 279 and 289 of the discussion may be confusing. While there is a description that the morning type is 15.1%, the night type is 7.8%, and the intermediate type is 77.1%, the results of the cross-section study show 18.1% of the students are the morning type and 0.9% are the night type, and the rest is an intermediate type. How did you derive the results of the Cross Section study? It was difficult to read from this paper, so please give me an additional explanation. The same thing is repeated twice in the paragraph, so please consider keeping it simple.

3)     About lines 416 to 418 in the conclusion. It is said that the factors for taking morning-style diurnal behavior are increasing. Did it increase compared to what?  Please add a clear explanation for this. There is also a statement that students' BMI values are increasing. What was this compared to? Please give a clear explanation. How does it compare to the fact that students are eating better? If you don't clarify the group to compare and the timing, I don't think you can make a comparative expression of that group. Please consider this point carefully.

Minor points

1) Careless mistakes are occasionally found, so it is recommended to calibrate properly. For example, put a space between 'thanthe' on line 153. Put a period after the 'increase' on line 208.

2) Isn't ‘α’ in line 117 ‘p’?

3) Although it is referred to as TFEQ-T in Tables 2 and 3, it is TFEQ in the footnote of the table, and it is recommended to unify it and add a clear explanation.

4) The explanation is redundant, and the same content is repeated meaninglessly. It is recommended to write in a simple sentence.

Author Response

We thank to the reviewer for his/her contribution to our manuscript. We will consider his/her comments in our future studies.

Dear Reviewer 2;

All requested corrections have been applied and painted in yellow. Furthermore, the entire text is "proofread" by the "native speaker".

  1. Major comments 1
    • The country's covid restrictions were totally removed during the time the research data were gathered, and students began getting lectures face-to-face. The covid period effect was therefore excluded from the evaluation because, in our opinion, it cannot be seen as a major impact.
  2. Major comments 2
    • The mentioned iterations have been corrected. Source of additional findings better presented; original article revised (references 23)
    • The repetitive sentences throughout the text were corrected; the sentences were expressed in a simpler way.
  3. Major comments 3
    • The mentioned sentence is repositioned in the discussion through explanation in other paragraphs of the text in order to provide a better understanding. Additionally, as the reviewer pointed out, this statement was removed since we cannot clearly comment on the lifestyle.
  4. Minor comments
    • Suggested fixes conducted.
    • All text has been reviewed; sentences have been simplified. The overall text has been made more understandable.

Round 2

Reviewer 1 Report

No further comments.

Author Response

Dear reviewer,

Thank you very much for your contributions to our article.

Best regards 

Reviewer 2 Report

It's a small point, but the sentence on lines 239-240 is a little strange, so please check it.

Author Response

Dear Reviewer,

First of all, I would like to thank you very much for your contributions to our manuscript. I made the revision you have requested and sending in the attachment.

Best regards
